# Impact of molecular symmetry on crystallization pathways in highly supersaturated KH$_2$PO$_4$ solutions

Yong Chan Cho [1], Sooheyong Lee[1,2], Lei Wang [1], Yun-Hee Lee[1], Seongheun Kim [3], Hyun-Hwi Lee[3], John Jonghyun Lee[4] & Geun Woo Lee [1,2] ✉

Solute structure and its evolution in supersaturated aqueous solutions are key clues to understand Ostwald's step rule. Here, we measure the structural evolution of solute molecules in highly supersaturated solutions of KH$_2$PO$_4$ (KDP) and NH$_4$H$_2$PO$_4$ (ADP) using a combination of electrostatic levitation and synchrotron X-ray scattering. The measurement reveals the existence of a solution-solution transition in KDP solution, caused by changing molecular symmetries and structural evolution of the solution with supersaturation. Moreover, we find that the molecular symmetry of H$_2$PO$_4^-$ impacts on phase selection. These findings manifest that molecular symmetry and its structural evolution can govern the crystallization pathways in aqueous solutions, explaining the microscopic origin of Ostwald's step rule.

More than 140 years ago, Ostwald[1] recognized that supersaturated solutions often preferred to transform into metastable intermediate phases (MIP) rather than thermodynamically stable one during crystallization, called Ostwald's step rule (OSR). Up to date, lots of such phenomena have been reported in many materials systems[2–21]. The OSR has been explained by a couple of hypotheses on solutions, i.e., kinetic competition of nucleation and growth rates depending on their crystal sizes[22], the minimizing entropy production[23], and the impact of solution structure on the formation of MIP[24]. Moreover, recent studies[2–21] have shown that the OSR is applicable to nucleation phenomenon itself. As an example, unlike a single-step nucleation (SSN) process by density fluctuation (i.e., simultaneous ordering of density and structure), the supersaturated solution can take a two-step nucleation (TSN) process, depending on the degree of supersaturation[25]; the densification forming dense liquid regions, pre-nucleation, or amorphous nanoparticles as the MIP in the supersaturated solution, and then structural fluctuation sequentially forming crystalline phases within the MIP. The TSN and SSN can be switched with the degree of supersaturation[25]. Moreover, if the separated fluctuation occurs with the various degree of densification and local orders of MIPs, the supersaturated solution can take the multiple pathways of nucleation or crystallization. The densification raises

fundamental questions for the causes of the OSR, such as the appearance and stability of the dense liquid region in early stage of the TSN and the mechanism of the phase selection in general. To address the causes of the OSR, the evolution of the solution structure (solute structure in particular) should be investigated as a function of supersaturation, which can decisively impact on the formation of the MIPs and the pathways of crystallization.

In this regard, a large effort has been devoted to measuring the solution structures as a function of supersaturation using various methods, such as Raman/IR spectroscopy[7,26–28], small-angle X-ray scattering (SAXS)[29–31], X-ray absorption fine structure study (XAFS)[32,33], time-resolved X-ray experiments[34], and neutron scattering[35]. Although these studies have revealed the important features of solute structures in undersaturation, the studies[33,35–39] have provided inconsistent results for molecular symmetries of anions, hydration structure, and the evolution of anionic molecules with PO$_4^{3-}$, H$_3$PO$_4$, H$_2$PO$_4^-$, and HPO$_4^{2-}$. Such inconsistent results are ascribed to the considerable contribution of background scattering from solvent atoms or molecules (neutron scattering[35], XAFS[32,33], SAXS[29]) at undersaturated conditions. If we can reduce the contribution of the solvent scattering by achieving high supersaturation, the solute structure will be revealed more clearly. However, the achievement of high supersaturation is not

[1]Frontier of Extreme Physics, Korea Research Institute of Standards and Science, Daejeon 34113, Republic of Korea. [2]Applied Measurement Science, University of Science and Technology, Daejeon 34113, Republic of Korea. [3]Pohang Accelerator Laboratory, POSTECH, Pohang 37673, Republic of Korea. [4]Department of Mechanical Engineering, Iowa State University, Ames, IA 50011, USA. ✉e-mail: gwlee@kriss.re.kr

trivial due to unavoidable heterogeneous nucleation from a container retaining the solution. Thus, the detection of the detailed solute structures and their evolution is still very challenging in supersaturated aqueous solutions where the MIP or dense liquid region form. In addition, a number of theoretical and numerical studies[2–4,7,25,40] have shown various results of the solute structure and its structural evolution at supersaturated conditions which are caused by the large uncertainties of the interactions between solute ions and solvent molecules.

Recent studies overcome the limitation and have achieved the highly supersaturated $KH_2PO_4$ (KDP) solution by using an electrostatic levitation (ESL) technique[13,15], which shows only a few solvent molecules per solute ion in the solution. Moreover, they directly observed that the supersaturated KDP solution took the multiple pathways of crystallization through an intermediate metastable crystalline phase, depending on the degree of supersaturation[13]. However, the origin of the OSR (or MIP) is still unanswered in microscopic viewpoint particularly, which requires the measurement of solute structure and its structural evolution.

In present study, we measure the solution structures of KDP and $NH_4H_2PO_4$ (ADP) in high supersaturation by the aid of a combination of ESL and synchrotron X-ray scattering, which gives almost one or two water molecules per ion at maximum supersaturation. The measurements of solutions structure and its structural evolution reveal that KDP solution shows a solution-solution transition with supersaturation, indicated by abrupt molecular symmetry change of the solute, while ADP solution does not show this behavior. This elaborates how KDP solution takes multiple pathways of crystallization, depending on the supersaturation[13]. The present work provides the impact of the solution structure on the formation of the metastable crystal phase based on the microscopic viewpoint of the Ostwald's step rule[1], which gives a way of studying the solute structure and its evolution in highly supersaturated solutions.

## Results

### Crystallization events of highly supersaturated solutions

By using the combination of ESL and in-situ synchrotron X-ray scattering devices, we can probe the concentration-dependent structural evolution of an aqueous solution up to high supersaturation (Fig. 1a, see also Methods section, Supplementary Figs. 2, 3, and Supplementary Movie 1). A levitated solution droplet continuously evaporates, resulting in increasing concentration up to about $S = 5.0$ (Here, supersaturation $S$ is given by the ratio of sample concentration ($C_s$) with respect to equilibrium concentration ($C_e$), i.e., $S = C_s/C_e$). We record the crystallization event of the solution as a function of supersaturation[40]. Figure 1b, c shows the probability distribution of the crystallization events in the levitated KDP and ADP solutions based on the statistical data accumulated from more than 158 times (for KDP) and 130 times (for ADP) experiments. Interestingly, the KDP solution exhibit two distinctive probability distribution of the crystallization events, which is consistent with the previous results[13], while ADP solution shows only one. In the KDP solution, the first probability distribution of the crystallization events at low supersaturation corresponds to the formation of a stable phase (tetragonal structure), while the later event at high supersaturation does the formation of a metastable phase (monoclinic structure) which transforms into the stable tetragonal phase later[13]. This reflects there are two different crystallization pathways depending on the level of supersaturation. We here study the impact of solution structure and its structural evolution on the crystallization pathways.

### Solute structure and evolution of highly supersaturated solutions

We measure the structure factor ($s(q)$) of the solutions up to $S = 4.1$ and 3.6 for KDP and ADP solutions, respectively (Fig. 2a, b, see also

Supplementary Note. 1). Both solutions show similar $s(q)$ and its evolution at undersaturation and low supersaturation. This may be caused by the common network of hydrogen-bonded $H_2PO_4^-$ molecules, since KDP and ADP crystals belong to the same crystal group ($\bar{4}2d$). However, as water evaporates (i.e., concentration increases), the structural evolution of two solutions is noticeably different (Fig. 2a, b); in case of KDP solution, a double-peak (Q1 and Q2) around 2 Å$^{-1}$ and 3 Å$^{-1}$ gradually decreases until $S = 3.2–3.3$, and then suddenly increases at higher supersaturation (see the arrows and the inset in Fig. 2a), which reflects a structural transition within the solution. In case of ADP solution, the double-peak continuously decreases and Q1 moves a little toward low q ranges with supersaturation marked by the black arrows in Fig. 2b. In addition, a small pre-peak appears around 1.2 Å$^{-1}$ (the blue arrow in Fig. 2b), which implies developing longer range ordering (i.e., hydrogen-bonded link of $H_2PO_4^-$) in the solution. The different structural evolution of the KDP and ADP solution means that the two solutions take different crystallization processes which can influence on their crystalline phase selection in Fig. 1b, c. It is worth emphasizing that the number of water molecules in the KDP and ADP solutions is about 2.6 and 1.0 per each ion on average at the highest supersaturation, respectively, which has never been experimentally achieved in both KDP and ADP bulk solutions in other reported experiments. Such environment with depleted water provides a great chance to detect the solute structure and its structural evolution during the network formation of $H_2PO_4^-$ ions in the solutions.

To scrutinize the detailed solute structures, we subtract water contribution (i.e., $O_w$ (oxygen of water)-$O_w$ scattering) from the total scattering of the solutions (see also Supplementary Note. 2 and Supplementary Tables 1, 2). The structure factors mainly contributed by the solutes (reduced structure factors, $\Delta s(q)$), are shown in Fig. 2c, d. As expected, the structure of both solutions looks very similar, in particular, at high $q$ range beyond 3 Å$^{-1}$ which is related to the short-range order (SRO) of the solute, since $H_2PO_4^-$ unit block prevails in both solutions. However, in the lower $q$ range, KDP solution clearly exhibits different $\Delta s(q)$ from ADP solution, and the difference becomes more distinct as concentration increases. In KDP solution, the intensity of the two peaks between 1 Å$^{-1}$ and 3 Å$^{-1}$ weakens up to $S = 3.2$ and they merge at around $S = 3.3$ and intensifies thereafter, while the two peaks become distinct with supersaturation in ADP solution. Thus, the solute structure and its structural evolution of KDP solution differ from those of ADP solution, although both solutions have the common $H_2PO_4^-$ unit blocks.

We deduce the topological information of the solution structures by analyzing the reduced pair distribution function (PDF) ($G(r) = 4\pi r$ ($\rho(r)-\rho_o$), where $\rho(r)$ is the number density of atoms, and $\rho_o$ is the average number density). In Fig. 3, the G1 peak at 1.54–1.55 Å should belong to P-$O_P$ distance (1.5–1.58 Å) of $H_2PO_4^-$ in the solution phase, which is consistent with previous studies[32,33,35,41,42]. As water evaporates, the contribution of P-$O_P$ pairs becomes dominant and thus the G1 peak gradually increases (Fig. 3a, b). Then, the intensity of the G1 peak suddenly decreases over $S = 3.2–3.3$ in KDP solution (Inset in Fig. 3a), while it continuously increases in ADP solution (inset in Fig. 3b). The sudden change of G1 peak in KDP solution becomes more distinct, when we remove the water contribution from total $G(r)$ (i.e., the reduced PDF, $\Delta G(r)$ in Fig. 3c, d). In addition, the coordination number of the first neighbor atoms ($n_{G1}$) decreases over $S = 3.2–3.3$ in KDP solution (Inset in Fig. 3a), but not in ADP solution (Inset in Fig. 3b). Considering the fact that the molecular symmetry of $H_2PO_4^-$ ions is $C_{2v}$[27,32,43], the decreasing intensity and the coordination number of the G1 peak reflect the formation of disordered or ill-defined P-$O_P$ distances within $H_2PO_4^-$ molecules in KDP solution.

In other words, the molecular symmetry of $H_2PO_4^-$ ions changes from $C_{2v}$ to lower symmetry. Interestingly, our previous study[13] has demonstrated that highly supersaturated KDP solution over $S = 3.2–3.3$ transforms into the metastable monoclinic KDP crystal

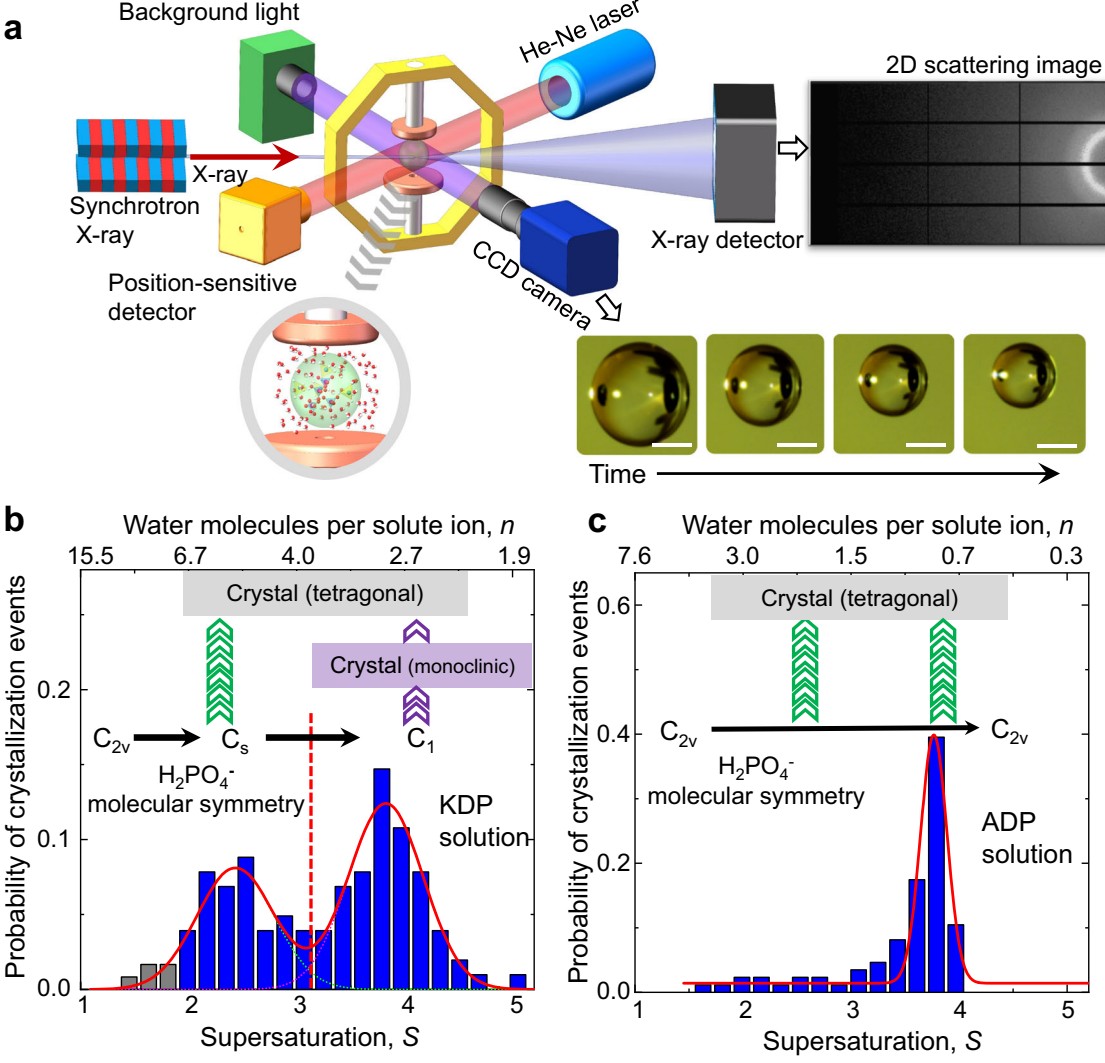

**Fig. 1 | In-situ real-time measurements of solution structure and the probability distribution of the crystallization events depending on supersaturation ($S$). a** Electrostatic levitation system to study solution structure by in-situ synchrotron X-ray scattering at Pohang Light Source. When a solution droplet reaches a targeted supersaturation value, the scattering signals are measured by using 2D X-ray detector. Time evolution of the levitated solution droplet is observed by using CCD cameras. The scale bar is 1 mm. Details of the device and its operational principles are elaborated in methods section and elsewhere[13,15,49] (see also Supplementary Figs. 2–4). Probability of crystallization events for KDP (**b**) and ADP (**c**) solutions depending on $S$. Number of water molecules per each ion ($n$) is presented on the upper-axis. Statistical data is accumulated from more than 158 times and 130 times

experiments for KDP and ADP solutions, respectively. Blue bars present the probability of crystallization events above S = 2.0. Gray bars of **b** indicate the crystallization at metastable zone width (MZSW) below $S$ = 2.0. Red lines are a cumulative curve with two Gaussian fitting curves (**b**) and with a Gaussian fitting curve (**c**) for blue bars. The $C_{2v}$, $C_s$, $C_1$ are corresponding to the molecular symmetry $H_2PO_4^-$ monomer and black arrows present the evolution of molecular symmetry with supersaturation. Green arrow indicates the crystallization to the directly stable tetragonal crystal (gray boxes in **b** and **c**) from the supersaturated solution. Purple arrow presents that the supersaturated solution transforms into monoclinic crystal (violet box in **b**) first, and then into tetragonal phase. Source data are provided as a Source Data file.

with four different tetrahedrons of $H_2PO_4^-$ molecules (i.e., $C_1$ molecular symmetry), while the solution at the lower saturation crystallizes directly into stable KDP crystal with a single type of the $H_2PO_4^-$ tetrahedron ($S_4$ molecular symmetry which is higher symmetry than $C_1$). In addition, in-situ Raman spectroscopy on supersaturated KDP solution clearly also indicates the intra-structure changes of the $H_2PO_4^-$ (see also Supplementary Fig. 5 for the details.); Raman spectra peak shape between 300 cm$^{-1}$ and 450 cm$^{-1}$, reflecting two in-plane P(OH)$_2$ bending modes, changes to asymmetric over S = 3.0 in KDP solution, while no change is observed in ADP solution. This is consistent with the change of G1 peak in X-ray result, displaying the change of intra-structure of the $H_2PO_4^-$ in KDP solution. Therefore, the present results provide the strong evidence that low concentration KDP solution (LCS) and high concentration KDP solution (HCS) have different molecular symmetries of $H_2PO_4^-$ ions.

Furthermore, this indicates that the molecular symmetry of the solutes can impact on the pathways of phase transition, facilitating the nucleation of metastable intermediate phase, and thus verifying the microscopic origin of Ostwald's step rule[22–24] in KDP and ADP solution system.

For the second nearest pairs, G2 peak is contributed by five different pairs (i.e., $O_P$-$O_P$ (≈2.49 Å and 2.57 Å)[35,41], $O_W$-$O_W$ (≈2.8 Å), $O_P$-$O_W$ (≈2.85 Å, and 2.77 Å,)[33,44], K-$O_p$ (≈2.8 Å in crystal phase) and K-$O_W$ (≈2.5 Å, 2.81 Å and 2.65 Å,)[35,45,46] pairs (N-$O_W$ ≈2.8 Å)[47] in case of ADP solution). As water continuously evaporates, the contribution of $O_W$-$O_W$, $O_P$-$O_W$, and K-$O_W$ pairs should decrease in the PDF of KDP solution, resulting in decreasing intensity and the shift toward shorter distance at 2.67 Å at $S$ = 4.1 (Fig. 3a). This indicates most contribution of G2 peak at high supersaturation comes from $O_P$-$O_P$ and K-$O_p$ (≈2.8 Å) pairs (NH$_4$-$O_p$ (≈2.9 Å) pairs for ADP solution). It should be noted that $O_P$-$O_P$

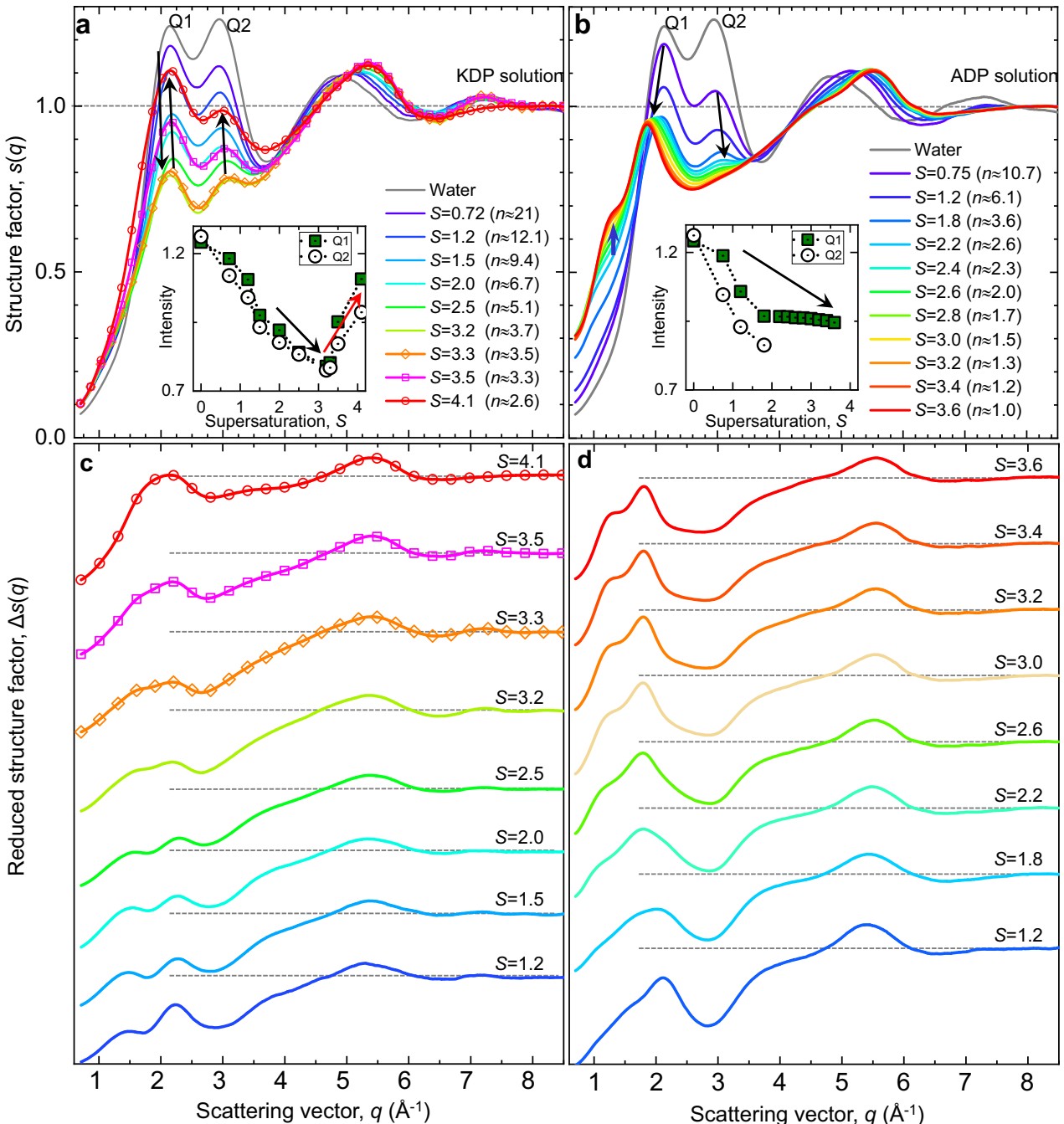

**Fig. 2 | The solution structure and its evolution in supersaturated aqueous solutions. a, b** X-ray structure factors ($s(q)$) of the levitated KDP and ADP solutions. The $s(q)$ above $S = 3.3$ of KDP solution are presented with the symbol and line together for clarity. Black arrows indicate the change of Q1 and Q2 peaks with increasing supersaturation. Blue arrow of **b** indicates a development of the peak around $q \approx 1.2 \, \text{Å}^{-1}$ in ADP solution. Black and red arrows of inset figures present the change of Q1 peak intensities with supersaturation. The $s(q)$ of the water droplet is shown with gray color for clarity. **c, d** Reduced structure factors ($\Delta s(q)$) of the solutes obtained by subtracting the water scattering contribution from total scattering intensity of the solutions based on the number ratio of water to total solution atoms ($a$) (See also Supplementary Note 2 and Supplementary Tables 1 and 2). The $\Delta s(q)$ above $S = 3.3$ of KDP solution are presented with the symbol and line together for clarity. $\Delta s(q)$ data is stacked with offset for visibility. Horizontal gray dashed lines represent the $\Delta s(q) = 1$ for offset results. Source data are provided as a Source Data file.

pairs are six per $H_2PO_4^-$ molecule, but K-$O_P$ pair is one. Therefore, the contribution of $O_P$-$O_P$ pairs is dominant for G2 peak.

The longer pair distance in PDF is related to the hydrogen-bonded link of $H_2PO_4^-$ ions, which is determined by inter-molecular pairing. Such medium range order (MRO) appears in the G3 and G4 peaks. While the exchange of G3 and G4 intensities is shown with increasing concentration in ADP solution (marked by black arrows in the inset figure of Fig. 3d), this tendency is relatively very weak in KDP solution.

This means that the hydrogen-bonded link process of $H_2PO_4^-$ molecules takes different ways in KDP and ADP solutions as supersaturation increases. The G3 peak in $\Delta G(r)$ is located at around 3.76 Å at the highest supersaturation in both solutions (Fig. 2c, d). Based on previous studies[32,33,35,36], the G3 peak may be contributed by P-$O_W$ (3.6–3.8 Å) in the first hydration, P-$O_{P'}$ (3.2–3.7 Å)[42] (here, $O_{P'}$ is oxygen in the neighboring $H_2PO_4^-$ in the dimeric structure [$H_2PO_4^-$]$_2$), and P-K ($\approx 3.64$ Å)[32] according to its crystalline phase (P-N $\approx 3.77$ Å) for ADP

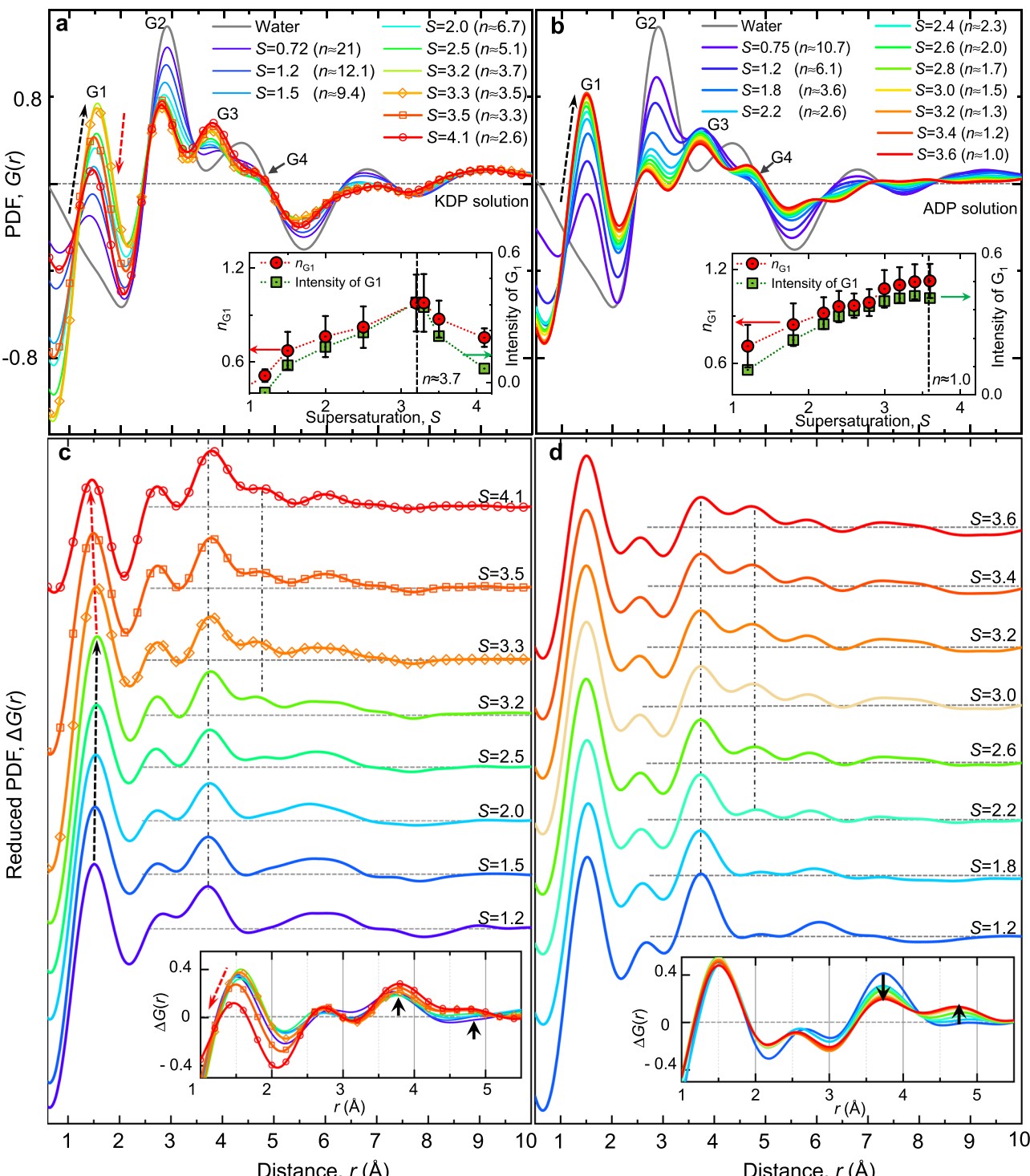

**Fig. 3 | Pair distribution functions (PDF, $G(r)$) of KDP and ADP solutions with supersaturation. a**, **b** $G(r)$ of the levitated KDP and ADP solutions with increasing supersaturation. The results above $S = 3.3$ of KDP solution are presented with the symbol and line together for clarity. G1 peak increases initially with supersaturation (black dashed arrow in **a**), and then decreases again after $S = 3.3$ (red dashed arrow in **a**) in KDP solution, while the G1 peak only continuously increases in ADP solution (black dashed arrow in **b**). Horizontal gray dashed line presents the $G(r) = 0$. Insets figures of **a**, **b** present the coordination number of the first nearest-neighbor atoms ($n_{G1}$) and intensity of G1 peak with supersaturation in KDP and ADP solutions. $n_{G1}$ is calculated with using four different integration methods which have been proposed in the structural study of liquid[50]. The $n_{G1}$ is determined by averaging the

obtained values obtained from four different integrations and the standard deviations are presented as error bar. **c**, **d** Reduced PDF for the solutes, $\Delta G(r)$, obtained by $\Delta s(q)$. Horizontal gray dashed lines present the $\Delta G(r) = 0$ for offset results. Black and Red dashed arrows of **c** present evolution of the first $\Delta G(r)$ peak below $S = 3.2$ and above $S = 3.3$, respectively. Black vertical dashed-dot lines are presented for eye-guide of the peak evolutions around $r = 3.7$ Å and $r = 4.8$ Å, which are also presented with thick black arrows in insets of **c** and **d**. Insets in **c**, **d** present the overlapped $\Delta G(r)$. Red dashed arrow of inset of **c** indicates the change of the first peak of $\Delta G(r)$ at highly supersaturated solutions. Source data are provided as a Source Data file.

solutions). Due to the small number of water molecules at the highest supersaturation, the prominent peak of G3 peak is mainly contributed by the P-O$_P$' and P-K(N) pairs. On the other hand, it is hard to identify the G4 peak in solution structure, although we may conjecture it with P-P distance between the connected H$_2$PO$_4^-$ ions. Since the distance of P-P pair is approximately 4.1–4.2 Å in their stable tetragonal crystalline phases, this pair may be relaxed in the solutions and expanded to the positions of the G4 peak at ≈4.62 Å, and ≈4.64 Å at the highest supersaturation in KDP and ADP solutions, respectively. From the position and shape of G3 and G4 peaks, we can recognize the formation of large size H$_2$PO$_4^-$ clusters with diameters greater than 1 nm in the solution, before crystal nucleation occurs. Since the cluster size is average value, we can imagine the existence of larger clusters than 1 nm in the solution which may reflect the early stage of two-step nucleation (i.e., dense liquid region)[15].

## Discussion

We found that the abrupt changes of the first peaks in structure factors and PDFs of the KDP solution, exhibiting changing SROs and MROs

with supersaturation. This is the strong signature of the solution-solution transition in KDP solution having two different molecular symmetries of H$_2$PO$_4^-$ ion with supersaturation. In addition, different structural evolution of the KDP and ADP solution means that H$_2$PO$_4^-$ ions take different network formations process with supersaturation. However, the molecular symmetry and MRO evolution have not been identified yet at such high supersaturation, which should be known to understand the solute structure of the two solutions. We here discuss the possible solute structures of KDP and ADP solutions with cluster modeling by using reverse Monte Carlo method (for details, see Supplementary Note 2).

A dimeric unit block of H$_2$PO$_4^-$ molecule has been suggested[28,37,38,43], as a representative conformation of the aggregates in the solutions. Such possibility is observed in the PDF (Fig. 3). That is, the distinct G3 and G4 peaks contributed by P-O$_P$' and P-P pairs should include two H$_2$PO$_4^-$ ions at least. In addition, a monomer of H$_2$PO$_4^-$ could not reproduce the peak position of Δ$s(q)$ for both solutions (marked by gray dash-dot line in Fig. 4c, d). Therefore, we examine all possible [H$_2$PO$_4^-$]$_2$ dimers which have been proposed in theoretical studies[38,43] (Fig. 4a). All [H$_2$PO$_4^-$]$_2$

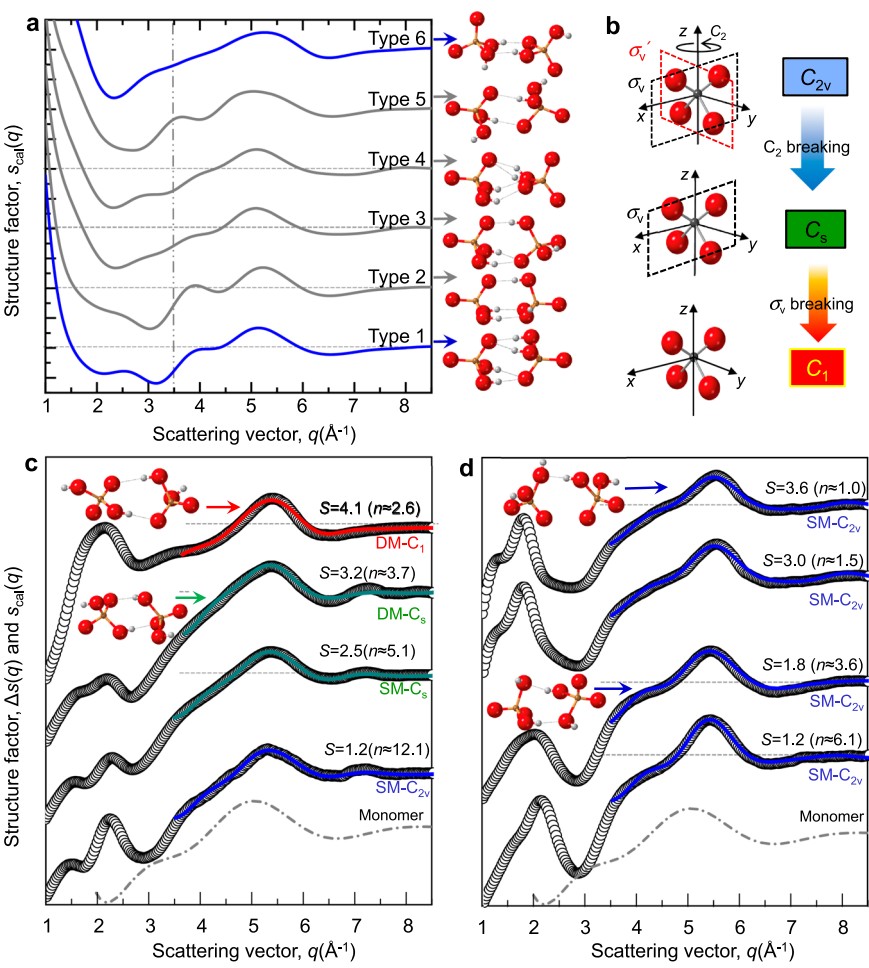

**Fig. 4 | The symmetry changes of H$_2$PO$_4^-$ based on the optimization of Δ$s(q)$ with [H$_2$PO$_4^-$]$_2$ dimeric cluster model. a** Calculated structure factor ($s_{cal}(q)$) of [H$_2$PO$_4^-$]$_2$ dimeric unit blocks. The $s_{cal}(q)$ are calculated from the [H$_2$PO$_4^-$]$_2$ dimeric cluster conformation types (Type 1–Type 6) suggested by theoretical studies[38,43]. Blue and gray arrows present the $s_{cal}(q)$ calculated from the corresponding to each dimeric cluster conformation type. Atomic coordinates and dimeric conformation types were obtained in refs. 38,43. **b** Symmetries of PO$_4$ with $C_{2v}$, $C_s$ and $C_1$. The $C_{2v}$ changes to $C_s$ which has only one mirror symmetry element ($\sigma_v$) by varying one of P-O$_P$ distances or/and one of O$_P$-P-O$_P$ angles. Breakage of the $\sigma_v$ symmetry results in $C_1$. **c, d** Experimentally obtained Δ$s(q)$ (open circles, Fig. 2c, d) and calculated $s_{cal}(q)$ (solid line) for dimeric model composed of two same (SM) or different (DM)

geometry monomers. All calculations are carried out from the same initial dimer model (Type 1 of **a**). Fitting is performed with Δ$s(q)$ data between $q$ = 3.5 Å$^{-1}$ and 8.5 Å$^{-1}$. Blue solid line is $s_{cal}(q)$ for the [H$_2$PO$_4^-$]$_2$ dimer with $C_{2v}$ symmetry (SM-$C_{2v}$). Green solid line presents dimer with $C_s$ symmetry (SM-$C_s$ and DM-$C_s$). Red solid line is $s_{cal}(q)$ for the dimer with $C_1$ symmetry (DM-$C_1$) (see also Supplementary Note 2 and Supplementary Tables 3 and 4). Here, hydrogen atoms are omitted in calculation. Gray dashed-dot lines are the $s_{cal}(q)$ of H$_2$PO$_4^-$ monomer. Horizontal gray dashed lines are presented for guiding $s(q)$ = 0. Dimer model figures of inset present the obtained conformation of [H$_2$PO$_4^-$]$_2$ from the indicated Δ$s(q)$ by red, green and blue arrows. Source data are provided as a Source Data file.

dimers similarly reproduce the characteristic feature observed over 3.5 Å$^{-1}$ in the experimental $\Delta s(q)$ (Fig. 4a). Here, we focus on the $\Delta s(q)$ over 3.5 Å$^{-1}$, since SRO of intra-cluster appears in high $q$ range. Among the [H$_2$PO$_4^-$]$_2$ dimeric clusters, we choose the type-1 cluster and relax it to fit $\Delta s(q)$, since the type-1 cluster has the lowest energy among the [H$_2$PO$_4^-$]$_2$ dimers[38].

While a dimer composed of two same monomers (SM) with $C_{2v}$ symmetry (SM-$C_{2v}$) reproduces the structure factors of solute in ADP solution, $\Delta s(q)$, through all concentrations in Fig. 4d, the symmetry of the dimer changes from $C_{2v}$ to $C_1$ with supersaturation in KDP solution; the $\Delta s(q)$ is reproduced by the dimer (SM-$C_{2v}$) for low concentration solution. Then, a dimer composed of same monomer with $C_s$ symmetry (SM-$C_s$) fits well the $\Delta s(q)$ in the range of $S = 1.5$–$2.5$. At $S = 3.2$, the dimer having two different monomers with $C_s$ symmetry (DM-$C_s$) give the better fitting. Finally, a dimer with DM-$C_1$ yields the best result at the highest supersaturation (see the detailed parameters for fitting in Supplementary Tables 3 and 4).

Our study for the evolution of solution structure can provide a crucial insight into the microscopic origin of the multiple pathways of nucleation or Ostwald's step rule in KDP and ADP solution system. According to van Santen[23], taking metastable intermediate phase(s) before transforming into stable phase yields minimum entropy production, which explains the Ostwald's step rule. This strongly indicates lower nucleation barrier for the metastable intermediate phase than for stable phase (i.e., $\Delta G_{ms}^* (\approx \sigma_{ms}^3/\Delta\mu_{ms}^2) < \Delta G_s^* (\approx \sigma_s^3/\Delta\mu_s^2)$, where $\sigma$ is the interfacial energy between the solution and the crystal phase, $\Delta\mu \approx \ln S$ (here, $S$ is supersaturation) is the driving force. Subscripts ms and s denote metastable and stable phases, respectively). Since the driving force for the nucleation of the metastable phase is generally smaller than that of the stable phase, the interfacial energy between the supersaturated solution and the metastable phase should be much smaller than that between the solution and the stable phase, i.e., $\sigma_{ms}^3 < \sigma_s^3 (\ln S_{ms}/\ln S_s)^2$. This expects the similar configurational structure of supersaturated solution with the intermediate metastable crystal phase. In previous work[13], we found that the metastable KDP crystal has monoclinic structure ($C_1c_1$) which is composed of four different tetrahedra of H$_2$PO$_4^-$ molecule with $C_1$ molecular symmetry, while the stable KDP crystal ($I\bar{4}2d$) has only one kind of tetrahedron with $S_4$ molecular symmetry (Fig. 4). The decreasing G1 peak and coordination number in KDP solution means that the molecular symmetry of H$_2$PO$_4^-$ ion with $C_{2v}$ changes toward lower symmetry. In addition, the cluster modeling shows that the symmetry of H$_2$PO$_4^-$ molecule changes from $C_{2v}$ to $C_1$ symmetry in KDP solution with supersaturation. Thus, the same molecular symmetry $C_1$ of the solute and the monoclinic crystal facilitates the nucleation of metastable intermediate phase. In short, the molecular symmetry ($C_1$) in SRO and structural evolution in MRO plays a key role as a basic template for the crystal building, which yields the lower crystal-liquid interfacial free energy for the metastable KDP crystal with $C_1c_1$ structure, but the higher interfacial free energy for stable KDP crystal with $I\bar{4}2d$ structure. This implies that crystallization occurs by particle attachment process (CPA)[2,4]; the ways of building block with the molecules is also a critical factor to determine the crystallization pathways.

In conclusion, we have measured the local structures of highly supersaturated aqueous solutions by using the combination of ESL and synchrotron X-ray diffraction. From the measured structure of the solutions, we discover the solution-solution transition, caused by changing the molecular symmetry of the solute from $C_{2v}$ to $C_1$ in the KDP solution. Our result suggests that the structural evolution of molecular structure in SRO and MRO ultimately determines the route of nucleation and phase selection, which has been hypothesized as the structural origin of Ostwald's step rule[1] in aqueous solution for a long time[23,24]. Therefore, the findings in KDP and ADP solutions provide an important way for atomic- and molecular-scale understanding of various nucleation phenomena that occurs through self-organization of

solutes in molecule-level and will impact a wide range of research areas from biology to material science[48].

## Methods

### Sample fabrication
Aqueous solutions are prepared by dissolving KDP of purity 99.99% (Aldrich INC) and ADP of purity 99.9% (Aldrich INC) in deionized water (resistivity 18.2 MΩ cm), respectively. To avoid the unwanted crystallization before the levitation, the initial concentration of solutions is fixed to $S = 0.72$ and $S = 0.75$ for KDP and ADP solution, respectively. For homogeneous mixture of solution, the solutions are stirred for 6 h, kept for 6 h at room temperature, and filtered with a pore size of 220 nm (BIOFIL). The filtered solution is finally delivered to a syringe equipped at bottom electrode in ESL. The values of pH are ≈4.19 and ≈3.81 for KDP and ADP solutions, respectively, reflecting that H$_2$PO$_4^-$ anions are dominant in the solutions, compared with other phosphate species.

### Levitation of solution droplet
The droplet is levitated at the position between top and bottom electrodes by using a voltage-position feedback system. A high voltage amplifier (Trek 10/10B-HS) provides a high electric voltage (1–5 kV, 300–500 Hz) to the top electrode. The He-Ne laser (1 mW of laser power with $\lambda = 633$ nm) incidents on the droplet with casting its shadow on a position sensitive detector (PSD, Hamamatsu Photonics C10443-03) to maintain the droplet position within ≈±20 μm. Temperature and relative humidity inside the sample chamber are maintained at 25.5 °C and 42 ± 2%, respectively.

### Determination of supersaturation
Supersaturation of the solution droplet is given by the ratio of sample concentration ($C_s$) to the equilibrium concentration of the solution ($C_e$) (i.e., $S = C_s/C_e$). Since solutes (KDP and ADP) are non-volatile and only water molecule evaporates, the concentration of the levitated solution droplet can be estimated from the measurement of droplet volume. In this study. the change of droplet volume is measured using a high-resolution B/W CCD camera every 5 s. We use a differential contrast method for the edge detection of the droplet image. The detected edge points in Cartesian coordinates are transformed into polar coordinates and then fitted with the 6$^{th}$-order Legendre polynomial, $(R(\theta) = \sum_{l=0}^{6} c_n P_1(\cos(\theta)))$, where $P_l(\cos(\theta))$ is the $l$-th order Legendre polynomial and $c_n$ is the coefficients determined by the 6th order Legendre polynomial fitting. The droplet volume is obtained by integrating the fitting results according to $V = (2\pi/3)\int_\pi^0 R(\theta)^3 \sin(\theta)d\theta$. Then the change of solution concentration can be obtained, which provides the evolution of supersaturation of solution droplet.

### Observation of crystallization event
In this study, it takes more than 4500 s for solution droplet to reach a maximum concentration value by water evaporation. In order to determine whether crystals formed or not, we monitored the surface and inside of the solution droplet by using the other CCD camera during the experiment. When the solution crystallizes at a supersaturation, we consider the supersaturation value of just before (≈5 s) the crystallization event (i.e., we did not use the solution images which include crystals). Since the evaporation rate is much slow, the change of supersaturation for 5 s is negligible. We carried out 158 and 130 times experiments for the crystallization of KDP and ADP solutions, respectively. Then, we get the statistics of crystallization events.

### X-ray scattering experiment and data acquisition
The ESL is installed at Pohang Light Source II (PLS II, 5A and 1C beamlines) for the in-situ synchrotron X-ray scattering experiment of

the levitated solution droplets. Monochromatized 18 KeV X-rays from a cryogenically cooled silicon (111) double crystal monochromator was delivered to the sample through a delivery pipe filed with helium gas. The X-ray detector (Pilatus-300 KW, $1475 \times 195$ pixels with a pixel size of $172\,\mu m \times 172\,\mu m$) was just placed behind the ESL chamber to collect the scattered signals from the solution droplet. When the supersaturation reaches the targeted value without any crystallization, X-ray beam is irradiated on droplet and 2D scattering image is recorded. The conversion of the 2D image to 1D data is carried out by using Fit2D program and Dioptas program. Subsequently, the $s(q)$ and $G(r)$ are obtained by using PDFgetx2 program.

### Raman scattering experiment and data acquisition
During the sample levitation, in-situ Raman measurement is performed in a transmission geometry with an incident laser ($\lambda = 532$ nm). The scattered beam is relayed via multiple high-quality silver mirrors to a Raman spectrometer (Dongwoo Optron, model DM500i). The delivered light is dispersed by a diffraction grating (1200 grooves $mm^{-1}$) and collected on a 2D detector (Andor DV401A-Bv) consisted of $1024 \times 24$ pixels (pixel size of $26\,\mu m^2$). In order to minimize the heating effect by laser, we use a typical laser power of 1.7 mW. At a given supersaturation, a single Raman spectrum was obtained by accumulating each spectrum every 3 s, with a total beam exposure time limited up to 21 s. All spectra at various supersaturation levels are measured independently.

### Reporting summary
Further information on research design is available in the Nature Portfolio Reporting Summary linked to this article.

## Data availability
The data that support the findings of this study are available from the corresponding author upon request. Source data are provided with this paper.

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

## Acknowledgements
The authors thank J. J. De Yoreo, P. G. Vekilov, H. Hwang, and Y. J. Kim for useful discussion and comments. This research was supported by the National Research Council of Science & Technology (NST) grant by the Korea government (MSIT) (No. CAP23041-100, G.W.L.), by the Converging Research Center Program through the Ministry of Science, ICT and Future Planning, Korea (Grants No. NRF-2014M3C1A8048818, and No. NRF2014M1A7A1A01030128, G.W.L.), by Characterization platform for advanced materials funded by Korea Research Institute of Standards and Science (KRISS-2022-GP2022-0013, G.W.L.) and partially by National Science Foundation of the United States (grant EPSCoR-2132131, J.J.L.).

## Author contributions
G.W.L. designed research; Y.C.C., S.L., L.W., Y.H.L., S.K., H.H.L. and J.J.L. performed experiment; Y.C.C and G.W.L. analyzed data; Y.C.C. and G.W.L. wrote the manuscript, which was commented by all authors.

## Competing interests
The authors declare no competing interests.
