## [Peer Review File · Nature Communications]

Impact of molecular symmetry on crystallization pathways in highly supersaturated KH_2PO_4 solutionsREVIEWER COMMENTS

Reviewer #1 (Remarks to the Author):

In this work, the authors describe the observation of the evolution of solutions of KH_2PO_4 (KDP) and $\text{NH}_4\text{H}_2\text{PO}_4$ (ADP) under conditions of high supersaturation. The elevated supersaturations are obtained using electrostatically levitated drops from which solvent evaporates over time. The lack of surfaces allows the system to reach supersaturations of up to $S \sim 5$. The authors monitor the evolution of the solute structure (the structure factor and corresponding pair distribution function) by means of synchrotron x-ray scattering. In each repetition of the experiment, the initial supersaturation was low and increased until crystallization occurs. From the statistics of over 100 repetitions, the frequency of crystallization as a function of supersaturation was obtained.

In general the results for the two molecules were quite different: in particular, the histogram of crystallization frequency versus supersaturation for the KDP shows two peaks whereas the ADP has only one. Differences in the structure and evolution of the peaks of the light scattering signal are interpreted, using MC simulations, as a change in symmetry of the KDP at higher supersaturations. This is promoted as an explanation of the KDP crystallizing via different nucleation pathways at higher and lower supersaturation and, ultimately, as explaining the microscopic mechanism underlying the Ostwald rule of steps for this system.

The experimental technique appears to be well-understood and reliable and the authors' interpretation of the results is convincing. The connection between the change in symmetry of the KDP molecule in solution and the eventual crystallization behaviour will be of interest to the broad community trying to understand the origins of polymorphism. However, there are several points that I feel should be addressed before the paper can be recommended for publication:

1. I do not understand how the supersaturation was determined at the moment of crystallization: can this be explained in the Methods or Supplementary Text?
2. I do not understand the definition of $G(r)$ in terms of the density given in line 140. First, I suppose the prefactor should be $4\pi r^2$? Second, the density $\rho(r)$ is, I believe, the conditional density that one observes if a molecule is fixed at the origin and not the unconstrained local density, which should be nearly constant. Third, I do not understand the subtraction of $\rho_{\{0\}}$ - this looks like the structure function and not the pair distribution function. (E.g., in liquids the structure function $h(r) = g(r) - 1$.) This is re-enforced by the fact that the $G(r)$ shown in the figure is sometimes negative and decays at long distances to zero, all of which are typical of the structure function and not the pair-distribution function.
3. On the basis of the evidence of these experiments, it is certainly reasonable to identify the structural changes of the KDP solute as the mechanism behind the OSR, it should be emphasized everywhere that this is proposed as the mechanism behind the OSR ****in this system**** and, certainly, not in general. Whether such structural changes of solutes are relevant in any other cases is an open question.

I recommend publication after the authors have addressed these issues.

Reviewer #2 (Remarks to the Author):

This paper investigates the structural evolution of highly supersaturated aqueous solutions of KH_2PO_4 (KDP) and $\text{NH}_4\text{H}_2\text{PO}_4$ (ADP). Through a combination of electrostatic levitation and synchrotron X-ray scattering, the study uncovers a solution-solution transition in KDP solutions popping up at increasing concentrations. The main findings of this paper suggest that molecular symmetry and its structural evolution play a crucial role in dictating crystallization pathways in aqueous solutions, offering insights into the microscopic mechanisms underlying Ostwald's step rule.

This insight is enabled by the experimental techniques implemented in the work, which, by taking advantage of acoustic levitation, enable the authors to investigate extremely high supersaturation conditions without observing nucleation initiated by heterogeneous surfaces.

The paper's findings are useful, as the microscopic behaviour of multicomponent liquid phases is challenging to probe experimentally.

In my view, however, the results fall short of the promise of explaining the Ostwald rule of stages, providing observations that enable rationalization but that are far from providing a quantitative description. Moreover, the clarity of the paper needs to be improved, and in my opinion, a number of points need to be addressed (see the following)

As such, while I think the results are interesting and publishable, in its current form, the paper is unsuitable for the general audience of Nat Comm. I recommend submitting it to a more specialized journal.

- The authors write:

"Moreover, recent studies²⁻²¹ have shown that the OSR is applicable to nucleation phenomenon itself. That is, unlike the classical nucleation theory describing a single-step nucleation process by density fluctuation, the supersaturated solution experiences a two-step nucleation (TSN) process which first shows densification in the supersaturated solution, yielding dense liquid regions, pre-nucleation, or amorphous nanoparticles as the MIP, and then structural fluctuation sequentially to form crystalline phases within the MIP."

This statement seems too general, multi-step nucleation processes and indeed non-classical nucleation are possible, but they are not necessarily always taking place.

Their importance strongly depends on the conditions in which nucleation takes place, with particular reference to the supersaturation of the parent phase.

As shown by simulation work (see i.e. Finney and Salvalaglio, WIREs 2023, and references therein) nucleation mechanism can switch from one to two steps depending on the driving force of the process, and the existence of complexes or adducts does not necessarily imply a two-step process is the most likely.

- The following sentence is unclear and should be rewritten:

"Thus, the separated fluctuation with densification and structural formation results in complicated, but abundant transition pathways of crystallization."

- The authors write:

"To address the causes of the OSR, the evolution of the solution structure (solute structure in particular) should be investigated as a function of supersaturation since the supersaturated solution is the last state just before the formation of MIP (i.e., dense liquid 40 region or metastable crystal)."

It is unclear what the authors mean by saying that the supersaturated solution is the "last state" before the formation of a liquid intermediate. In fact, the starting state of a crystallization process is typically a supersaturated solution. This is a matter of language but should be explained more clearly.

- When the authors cite "a number of theoretical and numerical studies^{2-4,7,39}", they fail to mention molecular simulation results showing how ubiquitous two-step processes are (see, for instance, Finney and Salvalaglio, WIREs 2023, and references therein)

- The fact that the levitated droplet continuously evaporates during experiments requires addressing the characteristic timescale associated with the formation of dense liquid phases in solution, compared with the characteristic timescale of evaporation. In other words, is it safe to assume that the pair

distribution functions reported in Fig 2 are obtained at constant S?

- What would be the implication of a varying S during the time devoted to collecting the $s(q)$ data?
- How did the author ensure that the timescales are sufficiently separate so that a $G(r)$ could be assigned a specific S value?

****Minor Comments****

The writing should be thoroughly revised to correct typos and improve readability throughout the paper.

Line 27 has been reported -> . have been reported

line 28 kinetical competition -> kinetic competition

line 66 - by calculation study -> by a computational study

Reviewer #3 (Remarks to the Author):

Cho et al's manuscript describes an investigation into the relationship between the conformation of solute molecules prior to crystallization and in the product crystal polymorph. This is achieved using a levitated droplet system where very high supersaturations can be reached prior to nucleation. Looking at the highly soluble crystals KDP and ADP, conditions are achieved where there are only a few water molecules per solute molecule prior to nucleation. Analysis of these systems using X-ray scattering and Raman spectroscopy reveals a relationship between the conformations in the solute and the product crystal polymorphs.

I am not an expert in X-ray scattering, so cannot really check all of the analysis for validity. However, I find the paper interesting, and if validated by experts in X-ray scattering, worthy of publication in Nature Communications.

I have a few comments:

(1) I find the final paragraph of the introduction rather long and overly-detailed. It is such a complete summary of the results and conclusions that it very much repeats the abstract and it hardly feels necessary to read the paper (as everything has been described..)

(2) P4 "Interestingly, the crystallization event of the KDP solution exhibits two distributions". I do not know what they mean by "distributions"

(3) P4. "The different structural evolution of the KDP and ADP solution means that the two solutions take different polymerization processes". I am not sure that polymerization is the best term. One does not usually describe crystallization as polymerization.

(4) P4. "It is worth to emphasize that the number of water molecules in the KDP and ADP solutions is about 2.6 and 1.0 per each ion on average at the highest supersaturation, respectively, which has never been experimentally obtained in either solution."

Do they mean never been observed during crystallization in bulk solution?

(5) Fig1. The legend is so long it is overwhelming. I would suggest splitting this into separate Figures.

(6) I find Figs 1b and 1c hard to decipher. What is the significance of the box that says "tetragonal crystal" over the top of of both graphs? What is the purple and green? And the purple and green

arrows?

(7) The Raman spectra provide an important validation of the X-ray data and I would recommend putting this data in the main paper.

(8) Figure 3. What do "types 1-6" refer to? Surely these should be described in the text/ shown in a Figure/ described in the legend. What are the colored dotted lines in Figs 3c and d?

(9) P11. "According to van Santen, taking metastable intermediate phase(s) before transforming into stable phase yields minimum entropy production".

I would have thought that it is most energetically favorable to maximize not minimize entropy.

(10) P11. " $\Delta\mu \sim \ln S$ is the driving force". Need to define S. Presumably this is supersaturation not entropy?

(11) Where possible, tables should be placed on one page in the SI. Tracked changes have also been left on in the SI and need to be removed.

Revision of NCOMMS-23-45733 for Nature Communications

We heartily appreciate the reviewers' comments which have improved our manuscript. We here appropriately answered for their questions and comments and revised the main manuscript and SI, which are marked by yellow color. We hope our reply satisfy reviewers.

Response to Reviewer #1

[Overall comment]: In this work, the authors describe the observation of the evolution of solutions of KH_2PO_4 (KDP) and $\text{NH}_4\text{H}_2\text{PO}_4$ (ADP) under condition is of high supersaturation. The elevated supersaturations are obtained using electrostatically levitated drops from which solvent evaporates over time. The lack of surfaces allows the system to reach supersaturations of up to $S \sim 5$. The authors monitor the evolution of the solute structure (the structure factor and corresponding pair distribution function) by means of synchrotron x-ray scattering. In each repetition of the experiment, the initial supersaturation was low and increased until crystallization occurs. From the statistics of over 100 repetitions, the frequency of crystallization as a function of supersaturation was obtained.

In general, the results for the two molecules were quite different: in particular, the histogram of crystallization frequency versus supersaturation for the KDP shows two peaks whereas the ADP has only one. Differences in the structure and evolution of the peaks of the light scattering signal are interpreted, using MC simulations, as a change in symmetry of the KDP at higher supersaturations. This is promoted as an explanation of the KDP crystallizing via different nucleation pathways at higher and lower supersaturation and, ultimately, as explaining the microscopic mechanism underlying the Oswald rule of steps for this system.

The experimental technique appears to be well-understood and reliable and the authors' interpretation of the results is convincing. The connection between the change in symmetry of the KDP molecule in solution and the eventual crystallization behaviour will be of interest to the broad community trying to understand the origins of polymorphism. However, there are several points that I feel should be addressed before the paper can be recommended for publication:

Response: We are grateful to for the Reviewer's in-depth evaluation of our study and for providing constructive feedback that has helped us improve our manuscript. Below, we provide point-by-point responses to Reviewer's concerns. We hope Reviewer finds our responses and revisions satisfactory.

[Comment-1]: I do not understand how the supersaturation was determined at the moment of crystallization: can this be explained in the Methods or Supplementary Text?

Response: Thank you for pointing out the determination of supersaturation. We believe that the

reviewer's question focuses on "at the moment of crystallization", but not the determination of the supersaturation itself. In this study, the concentration of the levitated solution droplet increases by evaporation and approaches to the highest supersaturation values for **more than 4500 seconds** (please see below **Fig. S1**). During this experiment, we take images of the levitated solution droplet **for every 5 seconds**. Moreover, the exposure time of X-ray is about 0.5 to 1 second which is good enough to get the diffraction signal in Synchrotron X-ray facility. Since the evaporation rate of the solution is much slower than measuring rates of X-ray diffraction and optical imaging, the supersaturation at the crystallization event can be considered to be the same as before 5 seconds. For the supersaturation calculation, we used solution images just before the crystallization (i.e., we did not use the solution images which include crystals).

If reviewer's question is related to how to determine the supersaturation, the determination of the supersaturation from a levitated solution droplet is well described in our previous studies [*PNAS* 2016(**113** (48), 13618-13623), *RSI* 2017(**88** (5), 055101), *Small* 2020 (**16** (11), 1907478), *Chem. Sci.* 2021(*Chemical Science* **12**, 179-187)]. According to reviewer's suggestion, we added the details in Supplementary Information, as below. (SI page 3)

(SI, Page3) "**Levitation of solution droplets:** ...In this study, the concentration of the levitated solution droplet increases by evaporation and approaches the highest supersaturation values for more than 4500 seconds. During this experiment, we take images of the levitated solution droplet for every 5 seconds. When the solution crystallizes at a supersaturation, we consider the supersaturation just before the crystallization event. Since evaporation rate is much slow, the change of supersaturation for 5 seconds is negligible.

Figure S1 The time dependence of the volume and supersaturation changes of the droplet.
a, The decrease of droplet volume depending on the levitation time. The dotted arrow indicates the occurrence of crystallization event. The droplet volume was obtained by the Legendre polynomial fitting from the detected droplet edge. **b,** The increase of supersaturation of droplet with increasing time. The dotted arrows indicate the maximal supersaturation points (S_{max}) just before the crystallization event. $S=1$ is presented for guiding the solubility limits of KDP and ADP solutions."

(SI, Page4) “**The determination of supersaturation value:** Supersaturation of the solution droplet is given by the ratio of sample concentration (c_s) to equilibrium concentration of the solution (C_e) (i.e., $S = C_s/C_e$). Therefore, we measure the concentration of the solution droplet during experiment. First, initial mass of solute and total volume of the solution are measured. Then, assuming that only water evaporates from the solution during experiment, we measure the volume of the solution droplet at a certain time t by using two CCD cameras in different direction (Figure S2). Thus, we can deduce the concentration of the sample at time t , during levitation experiment. The volume of the levitated solution can be calculated from the two-dimensional images of the solution droplet which are recorded by a high-resolution B/W CCD camera for every five seconds. In order to determine the radius of the solution droplet, we used a differential contrast method^{S4} which determines the edge detection of the shadow images. The detected edge points in Cartesian coordinates are transformed into polar coordinates and then fitted with the 6th-order Legendre polynomial, ($R(\theta) = \sum_{l=0}^6 c_n P_l(\cos(\theta))$), where $P_l(\cos(\theta))$ are the l -th order Legendre polynomial and c_n is the coefficients determined by the 6th order Legendre polynomial fitting (Fig.S2). The droplet volume (V) is obtained by integrating the fitting results according to $V = (2\pi/3) \int_{\pi}^0 R(\theta)^3 \sin(\theta) d\theta$. Finally, we can determine the concentration from the solute mass and solution volume, then the supersaturation is given from the definition, $S = C_s/C_e$.

Figure S2 The measurement of droplet volume. a, The schematic diagram for the volume measurement using B/W CCD camera and the droplet observation using the other CCD camera. b, The snapshot of droplet shadow image and the edge-detection. c, The volume calculation by applying the 6th -order Legendre polynomials fitting. “

[Comment-2]: I do not understand the definition of $G(r)$ in terms of the density given in line 140. First, I suppose the prefactor should be $4 \pi r^2$? Second, the density $\rho(r)$ is, I believe, the conditional density that one observes if a molecule is fixed at the origin and not the unconstrained local density, which should be nearly constant. Third, I do not understand the subtraction of $\rho_{\{0\}}$ - this looks the structure function and not the pair distribution function. (E.g., in liquids the structure function $h(r) = g(r) - 1$.) This is re-enforced by the fact that the $G(r)$ shown in the figure is sometimes negative and decays at long distances to zero, all of which are typical of the structure function and not the pair-distribution function.

Response: Thank you for the kind and precise comments. In general, pair distribution function (PDF) is denoted by $g(r)$ ($=\rho(r)/\rho_0$). Here, what we used is the reduced PDF ($G(r) = 4 \pi r (\rho(r) - \rho_0) = 4 \pi r \rho_0 (\rho(r)/\rho_0 - 1) = 4 \pi r \rho_0 (g(r) - 1)$). We used **the reduced PDF ($G(r)$)**, since $G(r)$ gives higher oscillation intensity than $g(r)$ in long distance, which distinctly shows changes in the oscillation (we note that RDF (radial distribution function= $4 \pi r^2 \rho_0 g(r)$) is often mentioned as PDF in simulation study.) To avoid misunderstanding, we correctly name it and give the definition in Supplementary Information.

[Comment-3]: On the basis of the evidence of these experiments, it is certainly reasonable to identify the structural changes of the KDP solute as the mechanism behind the OSR, it should be emphasized everywhere that this is proposed as the mechanism behind the OSR ****in this system**** and, certainly, not in general. Whether such structural changes of solutes are relevant in any other cases is an open question.

Response: We sincerely thank the reviewer for providing this valuable suggestion. We also agree that further studies should be done to generalize this issue for OSR. We revised the manuscript.

(On page 13, line 1) "Our study for the evolution of solution structure can provide a crucial insight into the microscopic origin of the multiple pathways of nucleation or Ostwald's step rule in KDP and ADP solution system."

[Comment-4] I recommend publication after the authors have addressed these issues.

Response: We thank Reviewer for this positive assessment of this work and for giving us the opportunity to further improve our study.

We hope Reviewer finds our responses and revisions satisfactory. We sincerely thank Reviewer again for providing this valuable suggestion and for taking the time to review our manuscript.

Response to Reviewer-2

[Overall comment]: This paper investigates the structural evolution of highly supersaturated aqueous solutions of KH_2PO_4 (KDP) and $\text{NH}_4\text{H}_2\text{PO}_4$ (ADP). Through a combination of electrostatic levitation and synchrotron X-ray scattering, the study uncovers a solution-solution transition in KDP solutions popping up at increasing concentrations. The main findings of this paper suggest that molecular symmetry and its structural evolution play a crucial role in dictating crystallization pathways in aqueous solutions, offering insights into the microscopic mechanisms underlying Ostwald's step rule.

This insight is enabled by the experimental techniques implemented in the work, which, by taking advantage of acoustic levitation, enable the authors to investigate extremely high supersaturation conditions without observing nucleation initiated by heterogeneous surfaces.

The paper's findings are useful, as the microscopic behaviour of multicomponent liquid phases is challenging to probe experimentally. In my view, however, the results fall short of the promise of explaining the Ostwald rule of stages, providing observations that enable rationalization but that are far from providing a quantitative description. Moreover, the clarity of the paper needs to be improved, and in my opinion, a number of points need to be addressed (see the following) As such, while I think the results are interesting and publishable, in its current form, the paper is unsuitable for the general audience of Nat Comm. I recommend submitting it to a more specialized journal.

[Response] We would like to thank the reviewers for their careful and constructive assessment and for giving us the opportunity to further improve our study. Although reviewer pointed out that our work is rational, but not 'quantitative' for supporting Ostwald's step rule (OSR) in microscopic perspective, we would appreciate it if reviewer consider following two points.

- 1) We guess that 'quantitative' pointed by reviewer2 means the size, formation rates, bond lengths or angles of local orders, time scale, nucleation barrier or activation energy, and so on, of the MIPs which can be provided by simulation or theoretical studies. In general, this quantitative information is hard to be directly measured in experiments and thus such effort has been very challenging. That's why most studies which is related to this issue have been performed by simulation. In experiment, a recent technique, liquid transmission electron microscopy (L-TEM) may give such information in nanometer scales [Loh, N. D. et al., *Nat. Chem.* **9**, 77-82 (2017), Habraken, W. J. E. M. et al. *Nat. Commun.* **4**, 1507 (2013)]. However, the microscopic method requires a very thin two-dimensional sample with a few nanometers thickness. This may give a spatial constraint for nucleation process, which may manipulate the nucleation process. Thus, same nucleation process may not be observed in bulk samples [Loh, N. D. et al., *Nat. Chem.* **9**, 77-82 (2017), De Yoreo, J. J. et al. *Science* **349**, aaa6760–aaa6769 (2015)]. In addition, the two-dimensional solution retained by a container in the microscopy is placed under heterogeneous condition for nucleation which can influence on the nucleation process as well as give small supersaturation. This may become serious obstacles to study the TSN in experiment, since the TSN might be expected to occur in high supersaturation according to

theoretical prediction as pointed out in the reference [Finney and Salvalaglio, *WIREs Comput Mol Sci.* e1697(2023)] by the reviewer.

In these regards, the present study using electrostatic levitation can provide great advantages, that is, highly supersaturated solution. What we achieved in this study is extremely high supersaturation, which gives **one or two water molecules per ion** in the **BULK** solution. This experimental condition can be comparable with the prediction of TSN process in the solutions in theory or simulation studies.

- 2) The hypotheses for the OSR have been suggested and tested for a long time, since Ostwald recognized the phenomenon more than 140 years ago. As we described in the manuscript, the hypotheses for the OSR of the solution are kinetic competition of nucleation and growth rates depending on their crystal sizes, the minimizing entropy production, and the impact of solution structure on the formation of MIP. Although these causes for OSR must be related to the evolution of solute structure with supersaturation, elaborate study for this purpose has been extremely challenging with highly supersaturated solution. In this regard, detecting the structural change of solute at the extremely high supersaturation provides concrete evidences that the molecular symmetry of the solute can impact on OSR in microscopic perspective, although our study may not perfectly satisfy reviewer2's request (i.e., 'quantitative'). In addition, the averaged structural information of the solute in bulk solution is quite representative under the high supersaturation so that can overcome or at least complement the limit of local area in microscopic study and the simulation study. Therefore, this work is very rare, challenging, and novel. We believe that our work will stimulate further insights for nucleation study to various research communities including chemistry, physics, pharmaceutical, materials, and so on. Therefore, we appreciate if the reviewer re-evaluates the value of our work.

According to the reviewer's comment, we have carefully revised our manuscript and supplementary Information. We hope Reviewer is satisfied with our responses and revisions.

[Comment-1]: The authors write: *"Moreover, recent studies²⁻²¹ have shown that the OSR is applicable to nucleation phenomenon itself. That is, unlike the classical nucleation theory describing a single-step nucleation process by density fluctuation, the supersaturated solution experiences a two-step nucleation (TSN) process which first shows densification in the supersaturated solution, yielding dense liquid regions, pre-nucleation, or amorphous nanoparticles as the MIP, and then structural fluctuation sequentially to form crystalline phases within the MIP."* This statement seems too general, multi-step nucleation processes and indeed non-classical nucleation are possible, but they are not necessarily always taking place. Their importance strongly depends on the conditions in which nucleation takes place, with particular reference to the supersaturation of the parent phase. As shown by simulation work (see i.e. Finney and Salvalaglio, *WIREs* 2023, and references therein) nucleation mechanism can switch from one to two steps depending on the driving force of the process, and the existence of

complexes or adducts does not necessarily imply a two-step process is the most likely.

[Response]: Thank you for the useful comments and we absolutely agree with reviewer. The TSN can occur in a supersaturated solution, not always, but depending on the degree of supersaturation. Since we have also found the evidence of two-step nucleation process with *highly supersaturated* NaCl solution by using electrostatic levitation providing containerless environment in experiment as well as simulation study [*Chem. Sci. 2021, 12, 179-187*], we absolutely agree with reviewer's comment. In addition, the reference [*Finney and Salvalaglio, WIREs Comput Mol Sci. e1697(2023)*] that reviewer mentioned cites our previous work with NaCl solution. Although we would like to just deliver the concept of TSN in the introduction part, we understand reviewer's concern about this issue which our writing may mislead readers. According to reviewer's comment, we revised the manuscript properly and added related references for readers.

(On page 2, line 8-13) "As an example, unlike a single-step nucleation (SSN) process by density fluctuation (i.e., simultaneous ordering of density and structure), the supersaturated solution can take a two-step nucleation (TSN) process, depending on the degree of supersaturation²⁵; the densification forming dense liquid regions, pre-nucleation, or amorphous nanoparticles as the MIP in the supersaturated solution, and then structural fluctuation sequentially forming crystalline phases within the MIP. The TSN and SSN can be switched with the degree of supersaturation²⁵"

²⁵Finney A. R. and Salvalaglio M., *Molecular simulation approaches to study crystal nucleation from solutions: Theoretical considerations and computational challenges*, *WIREs Comput. Mol. Sci.* e1697 (2023).

[Comment-2]: The following sentence is unclear and should be rewritten: "*Thus, the separated fluctuation with densification and structural formation results in complicated, but abundant transition pathways of crystallization.*"

[Response] : Thank you for this comment. We revised : this sentence more clearly in the manuscript:

(On page 2, line 13-15) "Moreover, if the separated fluctuation occurs with the various degree of densification and local orders of MIPs, the supersaturated solution can take the multiple pathways of nucleation or crystallization."

[Comment-3]: The authors write: "*To address the causes of the OSR, the evolution of the solution structure (solute structure in particular) should be investigated as a function of supersaturation since the supersaturated solution is the last state just before the formation of MIP (i.e., dense liquid region or metastable crystal).*" It is unclear what the authors mean by saying that the supersaturated solution is the "last state" before the formation of a liquid intermediate. In fact, the starting state of a crystallization process is typically a supersaturated solution. This is a matter of language but should be explained

more clearly.

[Response]: As referee's suggestion, we revised this sentence more clearly.

(page 2, line 17-19) *"To address the causes of the OSR, the evolution of the solution structure (solute structure in particular) should be investigated as a function of supersaturation, which can decisively impact on the formation of the MIPs and the pathways of crystallization."*

[Comment-4]: - When the authors cite "*a number of theoretical and numerical studies*^{2-4,7,39}", they fail to mention molecular simulation results showing how ubiquitous two-step processes are (see, for instance, Finney and Salvalaglio, WIREs 2023, and references therein)

[Response] → As Reviewer's kind indication, we added the references for the simulation works for the advanced achievements for two-step processes from solutions. We added the simulation papers for "a number of theoretical and numerical studies"^{2-4,25,40}

[Comment-5]: The fact that the levitated droplet continuously evaporates during experiments requires **addressing the characteristic timescale associated with the formation of dense liquid phases in solution, compared with the characteristic timescale of evaporation**. In other words, is it safe to assume that the pair distribution functions reported in Fig 2 are obtained at constant S?. -What would be the implication of a varying S during the time devoted to collecting the s(q) data? - How did the author ensure that the timescales are sufficiently separate so that a G(r) could be assigned a specific S value?

[Response]: Thank you for the detailed question. As we answered for Reviewer#1's question (1-1), the levitated solution droplet evaporates more than 4500 seconds, while our recording of sample images to deduce sample volume and concentration takes for every five seconds (See Fig S3 in SI). In the X-ray diffraction study, the highest supersaturation (S=4.1) is obtained after about 5000 seconds for KDP solution, while ADP solution approaches to the highest S(=3.9) after 6000 seconds. Since we got X-ray diffraction signal from the solution for 0.5 ~ 1 second typically, we think that **the timescale for supersaturation measurements is short enough to assume the constant S during X-ray diffraction measurement**.

We would like to emphasize that the determination of the supersaturation value in solution levitation experiment has been very carefully performed. By using a second CCD camera (See Fig S2), we monitored the surface and inside of the sample whether crystals formed or not during the experiment. Once we found any crystal, we discarded the sample and levitated a new solution droplet and restarted the experiment.

(SI, Page 3) **"Levitation of solution droplets** : ...*In this study, the concentration of the levitated solution droplet increases by evaporation and approaches the highest supersaturation values for*

more than 4500 seconds. During this experiment, we take images of the levitated solution droplet for every 5 seconds. When the solution crystallizes at a supersaturation, we consider the supersaturation just before the crystallization event. Since evaporation rate is much slow, the change of supersaturation for 5 seconds is negligible.

Figure S1 The time dependence of the volume and supersaturation changes of the droplet. **a**, The decrease of droplet volume depending on the levitation time. The dotted arrow indicates the occurrence of crystallization event. The droplet volume was obtained by the Legendre polynomial fitting from the detected droplet edge. **b**, The increase of supersaturation of droplet with increasing time. The dotted arrows indicate the maximal supersaturation points (S_{max}) just before the crystallization event. $S=1$ is presented for guiding the solubility limits of KDP and ADP solutions”

[Minor Comment]: The writing should be thoroughly revised to correct typos and improve readability throughout the paper.

Line 27 has been reported -> have been reported

line 28 kinetical competition -> kinetic competition

line 66 - by calculation study -> by a computational study

[Response]: Thanks for the kind indications, we revised this sentence more clearly and revised to correct typos and improve readability throughout the paper.

We sincerely thank Reviewer again for providing these valuable suggestions. We consider that experimental measurements of solution structure evolution using X-ray scattering and the result for the relation between the molecular symmetry change and eventual crystal structure will be of interest to the broad scientific research community trying to understand the crystallization aqueous solution in highly

supersaturated state. We hope that Reviewer is satisfied with our reply and consider the suitability of the manuscript for Nature Communications.

We sincerely thank Reviewer again for taking the time to review our manuscript.

Response to Reviewer #3

[Overall comment]: Cho et al's manuscript describes an investigation into the relationship between the conformation of solute molecules prior to crystallization and in the product crystal polymorph. This is achieved using a levitated droplet system where very high supersaturations can be reached prior to nucleation. Looking at the highly soluble crystals KDP and ADP, conditions are achieved where there are only a few water molecules per solute molecule prior to nucleation. Analysis of these systems using X-ray scattering and Raman spectroscopy reveals a relationship between the conformations in the solute and the product crystal polymorphs. I am not an expert in X-ray scattering, so cannot really check all of the analysis for validity. However, I **find the paper interesting, and if validated by experts in X-ray scattering, worthy of publication in Nature Communications.** I have a few comments:

[Response]: We thank Reviewer for the careful and positive assessment of our work and for giving us the opportunity to further improve our study. We are grateful to Reviewer's in-depth evaluation of our study and for providing positive feedback that has helped us improve our manuscript. Below, we provide point-by-point responses to Reviewer's comments. We hope Reviewer finds our responses and revisions satisfactory.

[Comment-1]: I find the final paragraph of the introduction rather long and overly-detailed. It is such a complete summary of the results and conclusions that it very much repeats the abstract and it hardly feels necessary to read the paper (as everything has been described..)

[Response]: We thank Reviewer for much valuable comment. According to Reviewer's comment, we revised the final paragraph of the introduction for pointing out the significance of this work.

*(On page3 line 14-23), "In present study, we measure the solution structures of KH_2PO_4 (KDP) and $NH_4H_2PO_4$ (ADP) in extremely high supersaturation by the aid of the-state-of-the-art, a combination of electrostatic levitation (ESL) and synchrotron X-ray scattering, which gives almost **one or two water molecules per ion** at maximum supersaturation. The measurements of*

solutions structure and its structural evolution reveal that KDP solution shows a solution-solution transition with supersaturation, indicated by abrupt molecular symmetry change of the solute, while ADP solution does not show this behavior. This elaborates how KDP solution takes multiple pathways of crystallization, depending on the supersaturation¹³. The present work provides the impact of the solution structure on the formation of the metastable crystal phase based on the microscopic viewpoint of the Ostwald's step rule²⁰, which paves a new way of studying the solute structure and its evolution in highly supersaturated solutions.

[Comment-2]: (2) P4 “Interestingly, the crystallization event of the KDP solution exhibits two distributions”. I do not know what they mean by “distributions”

[Response]: Thank you for much helpful comment, Crystal nucleation statistically occurs through density fluctuation. Thus, we find that the levitated droplet crystallizes over a wide range of supersaturation, which shows tetragonal phase and monoclinic phase at low and high supersaturation, respectively in KDP solution. Each crystallization exhibits a broad distribution of the event due to the statistical nature of the nucleation. Therefore, we meant that the distribution is of the probability of crystallization events. We fit the probability of crystallization events by a Gaussian function. Unlike the probability of crystallization events in ADP solution, it in KDP solution is fitted by two Gaussian peaks. We revised the manuscript more clearly in the first paragraph on page4.

(On page 5, line 9-16): *“Figure 1b and 1c show the probability distribution of the crystallization events in the levitated KDP and ADP solutions based on the statistical data accumulated from more than 150 times (for KDP) and 130 times (for ADP) experiments. Interestingly, the KDP solution exhibit two distinctive probability distribution of the crystallization events, which is consistent with the previous results¹³, while ADP solution shows only one. In the KDP solution, the first probability distribution of the crystallization events at low supersaturation corresponds to the formation of a stable phase (tetragonal structure), while the later event at high supersaturation does the formation of a metastable phase (monoclinic structure) which transforms into the stable tetragonal phase later¹³.”*

[Comment-3]: P4. “The different structural evolution of the KDP and ADP solution means that the two solutions take different polymerization processes”. I am not sure that polymerization is the best term. One does not usually describe crystallization as polymerization.

[Response]: Thank you for the important comment. Here, polymerization does not mean crystallization, since we study the structural evolution of solution. In fact, H_2PO_4^- ion makes clusters from monomer, dimer, tetramer, ..., as supersaturation increases. We called this process the polymerization. Through the literature survey, many researchers have used this terminology. However, for clarity, we put an explanation for this with “hydrogen-bonded link of H_2PO_4^- ” in manuscript.

[Comment-4]: P4. “It is worth to emphasize that the number of water molecules in the KDP and ADP solutions is about 2.6 and 1.0 per each ion on average at the highest supersaturation, respectively, which has never been experimentally obtained in either solution.” Do they mean never been observed during crystallization in bulk solution?

[Response]: We meant that such high supersaturation level has never been achieved on the KDP and ADP bulk solution using container methods.” This is not related to crystallization, but only bulk solution just before the crystallization event occurs at the highest supersaturation.

(page 6, line 9-10) “.. *which has never been experimentally achieved in both KDP and ADP bulk solutions in other reported experiments.*”

[Comment-5]: (5) Fig1. The legend is so long it is overwhelming. I would suggest splitting this into separate Figures.

[Response]: According to the helpful comment, we split Figure 1 into separate figures (Figure1 and Figure 2)

[Comment-6] (6) I find Figs 1b and 1c hard to decipher. What is the significance of the box that says “tetragonal crystal” over the top of both graphs? What is the purple and green? And the purple and green arrows?

[Response]: Thank you pointing out it, we intended to show different crystallization pathways, depending on supersaturation level. The “tetragonal crystal” means a final structural phase to present KDP and ADP crystals have ‘*tetragonal structure*’ at room temperature. The green and purple color present the supersaturation regions which form tetragonal and monoclinic crystals, respectively. The green arrow indicates the crystallization to stable tetragonal crystal (the grey box in Fig 1 **b,c**) from the supersaturated solution, while purple arrow shows that the supersaturated solution transforms into monoclinic crystal (the violet box in Fig. 1 **b**) first, and then into tetragonal phase. According to Reviewer’ comment, we added the detailed explanation for the Fig.1 cations.

(Cation for Fig.1) “...*The green and purple color present the supersaturation regions which form tetragonal and monoclinic crystals, respectively. The green arrow indicates the crystallization to the stable tetragonal crystal (the grey box in Fig 1 **b,c**) from the supersaturated solution, while purple arrow shows that the supersaturated solution transforms into monoclinic crystal (the violet box in Fig. 1 **b**) first, and then into tetragonal phase.*”

[Comment-7]: The Raman spectra provide an important validation of the X-ray data and I would

recommend putting this data in the main paper.

[Response]: We appreciate the reviewer's constructive suggestion which can improve the strength of our manuscript. According to Reviewer's suggestions, we tried to put the data into the main manuscript. However, we could not find proper position of this information. Although the Raman spectra reflects the change of bending mode of the P(OH)₂ molecule, indicating the change of molecular configuration in the nearest atom (i.e., intra-molecule), we feel that the information is the same as that from X-ray data. Moreover, we feel that putting all information and explanation of the Raman spectra into the paragraph of X-ray scattering analysis may disrupt the reading flow for audience with PDF analysis. Therefore, we would like to keep the figure of Raman spectra in SI and, instead, put a little more detailed information in the main text, although we agree and thank the reviewer's suggestion and purpose. We hope that reviewer consider this practical situation.

(page 8, line 14-18): "In addition, in-situ Raman spectroscopy on supersaturated KDP solution clearly reveals the intra-structure changes of the H₂PO₄⁻ (see Supplementary Section-1 for the details.); Raman spectra peak shape between 300 cm⁻¹ ~ 450 cm⁻¹, reflecting two in-plane P(OH)₂ bending modes, changes to asymmetric over S=3.0 in KDP solution, while no change is observed in ADP solution. This is consistent with the change of G1 peak in X-ray result, displaying the change of intra-structure of the H₂PO₄⁻ in KDP solution."

(page 15, line 6-13 in Method) "Raman scattering experiment and data acquisition. During the sample levitation, in-situ Raman measurement is performed in a transmission geometry with an incident laser with wavelength of 532 nm. The scattered beam is relayed via multiple high-quality silver mirrors to a Raman spectrometer (DONGWOO OPTRON; model DM500i). The delivered light is dispersed by a diffraction grating (1,200 Gr/mm) and collected on a 2D detector (Andor DV401A-Bv) consisted of 1,024 × 24 pixels (pixel size of 26 μm²). In order to minimize the heating effect by laser, we use a typical laser power of 1.7 mW. At a given supersaturation, a single Raman spectra was obtained by accumulating each spectrum every three seconds, with a total beam exposure time limited up to 21 seconds. All spectra at various supersaturation levels were measured independently."

[Comment-8]: Figure 3. What do "types 1-6" refer to? Surely these should be described in the text/ shown in a Figure/ described in the legend. What are the colored dotted lines in Figs 3c and d?

[Response]: The "types 1-6" indicated the conformation types of [H₂PO₄]₂ dimeric cluster, corresponding to the neighboring the dimer figures in Fig.3. The color dotted lines are presented for guiding $s(q) = 0$, in every corresponding $s_{cal}(q)$. According to Reviewer's comment, we added the more detail descriptions for the "types 1-6" and the colored dotted lines.

(Caption for Fig.4): "...The presented $s_{cal}(q)$ are calculated from the [H₂PO₄]₂ dimeric cluster conformation types (type-1~type-6) suggested by theoretical studies^{37,40}(right panel)." And "The

color dotted lines are presented for guiding $s(q) = 0$, in every corresponding $s(q)$.

[Comment-9]: P11. “According to van Santen, taking metastable intermediate phase(s) before transforming into stable phase yields minimum entropy production”. I would have thought that it is most energetically favorable to maximize not minimize entropy.

[Response]: We carefully checked a couple of papers and the original paper of Santen again. According to the R. A. van Santen's study based on irreversible thermodynamics [*J. Phys. Chem.* 1984,88, 5768-5769], “If a reaction can result in several products, it is not the stablest state with the least amount of free energy that is initially obtained, but the least stable one, lying nearest to the original state in free energy”, and “It is shown that Ostwald’s step rule minimizes entropy production”. Since the metastable intermediate phase is not the most stable state, the formation of intermediate phase from the supersaturated solution means that the MIP has energetically similar state with the supersaturated solution, compared with final stable phase, which minimizes the entropy production, if the reaction rate of the MIP is similar to that of the final stable phase.

[Comment-10]: P11. “ $\Delta\mu \sim \ln S$ is the driving force”. Need to define S. Presumably this is supersaturation not entropy?

[Response]: *The supersaturation S is given by the ratio of sample concentration (C_s) with respect to equilibrium concentration (C_e), i.e., $S = C_s/C_e$.) at line 82. In order to avoid a misunderstanding of reader, we denoted the definition S with $\Delta\mu \sim \ln S$ for readability.*

(On page 13, line 6) “where $\Delta\mu \sim \ln S$ (here, S is supersaturation)”

[Comment-11]: Where possible, tables should be placed on one page in the SI. Tracked changes have also been left on in the SI and need to be removed.

[Response]: Thank you for the very kind comments. According to Reviewer’s comment, we reedited the supplementary information including its tables more suitably. And we remove the left tracked changes, that had been contained as mistakes.

We sincerely thank Reviewer again for providing this valuable suggestion and for taking the time to review our manuscript.

REVIEWERS' COMMENTS

Reviewer #1 (Remarks to the Author):

I have reviewed the revised manuscript and the authors' responses to my comments. I believe that they have answered everything satisfactorily. I recommend publication of the paper in its present form.

Reviewer #2 (Remarks to the Author):

I am satisfied with the answers provided by the authors and with their substantial work on the paper. I am happy to recommend the publication of this work in its current form.